# Differentiable Spike: Rethinking Gradient-Descent for Training Spiking Neural Networks

**Yuhang Li**[1]*, **Yufei Guo**[2]*, **Shanghang Zhang**[3], **Shikuang Deng**[1], **Yongqing Hai**[2], **Shi Gu**[1]✉

[1]University of Electronic Science and Technology of China,
[2]Peking University, [3]University of California Berkeley
liyuhang699@gmail.com, yfguo@pku.edu.cn, gus@uestc.edu.cn

## Abstract

Spiking neural networks (SNNs) have emerged as a biology-inspired method mimicking the spiking nature of brain neurons. This biomimicry derives SNNs' energy efficiency of inference on neuromorphic hardware. However, it also causes an intrinsic disadvantage in training high-performing SNNs from scratch since the discrete spike prohibits the gradient calculation. To overcome this issue, the surrogate gradient (SG) approach has been proposed as a continuous relaxation. Yet the heuristic choice of SG leaves it vacant how the SG benefits the SNN training. In this work, we first theoretically study the gradient descent problem in SNN training and introduce finite difference gradient to quantitatively analyze the training behavior of SNN. Based on the introduced finite difference gradient, we propose a new family of Differentiable Spike (Dspike) functions that can adaptively evolve during training to find the optimal shape and smoothness for gradient estimation. Extensive experiments over several popular network structures show that training SNN with Dspike consistently outperforms the state-of-the-art training methods. For example, on the CIFAR10-DVS classification task, we can train a spiking ResNet-18 and achieve **75.4**% top-1 accuracy with 10 time steps.

## 1 Introduction

Recently, spiking neural networks (SNNs) have received increasing attention due to their biology-inspired neural behavior and efficient computation. SNNs deal with binary spike information and therefore enjoy the advantage of multiplication-free inference. Neuromorphic hardware such as TrueNorth [1] and Loihi [2] demonstrates that SNNs can save energy by orders of magnitude. Hybrid architecture like Tianjic [3] suggests its potential power for general intelligence when integrated with traditional artificial infrastructure. However, the bio-mimicry also causes an intrinsic disadvantage in training high-performing SNNs from scratch due to the discrete spike. As a result, obtaining a high-performing SNN has long been a critical problem limiting SNN's deployment in practice.

To be more specific, the appealing advantage of the multiplication-free computation in SNN is not recognized in its training [4]. Modern ML frameworks such as Pytorch [5], JAX [6], and Tensorflow [7] do not provide efficient and general instructions or high-level functions to accelerate the convolution with 0/1 spike activation. Moreover, the binary activation in SNN produces all-or-nothing gradients [8] that are incompatible with typical gradient-based optimization methods for efficient neural network training. To circumvent this problem, many works adopt *surrogate gradients* [8] (typically the derivative of a soft-relaxed function) to replace the spike non-linear gradient. Although the heuristic choice of SG is beneficial to enable the gradient descent in SNN training, the theoretical soundness of SG has not been justified so far. What's more, the gradient descent in SNN training may follow different updating rules from ANN. In fact, since the true gradient

---

*Equal Contributions. ✉Corresponding author. Primarily supported by NSF-61876032.

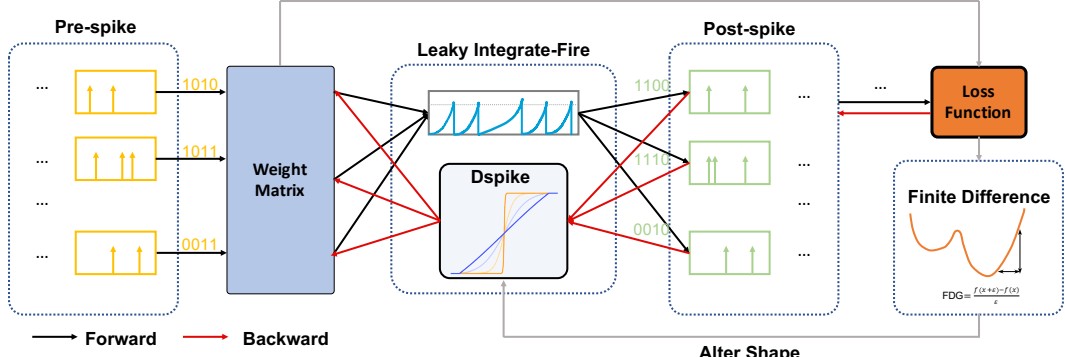

**Figure 1:** Overall workflow of the proposed algorithm. We use a general, flexible Dspike function and control its shape with finite difference method to facilitate the surrogate gradient descent in training SNN.

value of spike activation does not exist, the adoption of SG raises a new question of how to replace the $\infty$-gradient adaptively through the training process for a good updating strategy of parameters.

In this work, we study this specialized problem of adaptive gradient estimation in SNN training. We first revisit the gradient descent algorithm with the finite difference method to accurately depict the loss landscape of adopting a surrogate gradient for the non-differentiable spike firing function. Then we further propose the Differentiable Spike (Dspike) function, which can adaptively change its shape and capture the direction of finite difference gradients. With such adaptive surrogate gradients, we establish an efficient algorithm of training SNN with large-scale models. The overall workflow of our algorithm is visualized in Fig. 1.

Our contributions can be summarized as follows:

- We identify the problem of adaptively estimating gradient in SNN training and propose a novel analysis framework for SNN's surrogate gradient, through which we are able to not only explain why it works well but also indicate how we can improve its adaptivity to data through training.
- We also introduce the novel Dspike function built from the hyperbolic tangent function. The Dspike function can learn its shape accordingly by matching it with the finite difference gradient.
- Extensive experiments on both static and dynamic datasets show that our method is highly effective and efficient. For example, our model can achieve **75.4**% top-1 accuracy on the CIFAR-DVS. Our SNN also only consumes **7.8**% energy of the corresponding ANN on CIFAR-10.

## 2   Related Work

**ANN-SNN Conversion.**   Generally, the algorithms to obtain a high-performance can be divided into two routes: (1) ANN-SNN conversion and (2) direct training SNN from scratch. Conversion-based methods [9, 10] utilize the knowledge from ANN and convert the ReLU activation to spike activation mechanism. This type of method can produce an SNN in a short time. For example, Rueckauer *et al*. [11] propose to traverse the training dataset to find the maximum activation. Then, the SNN can be converted by directly replacing several modules. In [12], maximum activation is abandoned since it may out-lie the central range of activation. The authors choose to find the percentile numbers for threshold voltage to improve the robustness. TSC [13] adopts the soft-reset (also known as reset by subtraction) mechanism and no leakage for conversion, which proves to be effective in conversion. Deng & Gu [14] decompose the conversion error to each layer, then propose to reduce the error by shifting the bias term. [15] first propose the calibration to reduce the conversion accuracy. ANN-SNN conversion is fast and accurate. However, achieving near lossless conversion requires a considerable amount of time steps ($>$200) to accumulate the spikes, which may significantly add up the latency.

**Direct Training SNN.**   In contrast to the conversion method, direct training of SNNs can lower the time step and may further reduce energy consumption. Direct training leverages gradient-descent algorithms.  And many works have been developed to tackle the non-differentiability of spike activation. This non-differentiable activation function is similar to the quantization or binarization

neural network [16, 17, 18] or adder neural network [19]. [20] achieve high performance on CIFAR-10 and other neuromorphic datasets. Some works use probabilistic models to approximate the gradient [21], which is also developed in binary stochastic model [22]. Another common approach is to use rate-encoding networks for gradients calculation [23, 24]. They quantize the spike rate as the magnitude of information transmitted. (Wu *et al.* 2019 [25] improves the leaky integrate-and-fire (LIF) model to an iterative LIF model and develops STBP learning algorithm. Zheng *et al.* [26] further propose the tdBN algorithm to balance the gradient norm and smooth the loss function in SNN training, enabling large-scale model optimization.

# 3 Preliminary

## 3.1 Leaky Integrate-and-Fire Model

We use bold italic letters for vector representations. For example, $\boldsymbol{x}$ and $\boldsymbol{y}$ denote the input and target output variables. Bold capital letters like $\mathbf{W}$ denote the matrices (or tensors as clear from the text). Constants are denoted by small upright letters, e.g., $a$. With bracketed superscript and subscript, we can denote the time dimension and the element indices, respectively. For example, $\boldsymbol{u}_i^{(t)}$ means the $i$-th membrane potential at time step $t$. We adopt the well-known Leaky Integrate-and-Fire (LIF) neuron model for a spiking neural network (SNN). Given a membrane potential $\boldsymbol{u}^{(t)}$, the pre-synaptic potential is updated as

$$\boldsymbol{u}^{(t+1),\text{pre}} = \tau \boldsymbol{u}^{(t)} + \boldsymbol{c}^{(t+1)}, \text{where } \boldsymbol{c}^{(t+1)} = \mathbf{W} \boldsymbol{x}^{(t+1)}. \tag{1}$$

Here, $\tau$ is a constant leaky factor within $(0, 1)$ and $\boldsymbol{c}^{(t+1)}$ is the pre-synaptic input at time step $t + 1$. The input can be calculated from the dot-product between the weights and the spike from the previous layer. Let $V_{th}$ be the firing threshold, the membrane potential will fire a spike when it exceeds the threshold, and then hard-reset to 0, given by

$$\boldsymbol{y}^{(t+1)} = \begin{cases} 1 & \text{if } \boldsymbol{u}^{(t+1),\text{pre}} > V_{th} \\ 0 & \text{otherwise} \end{cases}, \quad \boldsymbol{u}^{(t+1)} = \boldsymbol{u}^{(t+1),\text{pre}} \cdot (1 - \boldsymbol{y}^{(t+1)}) \tag{2}$$

After firing, the spike output $\boldsymbol{y}^{(t+1)}$ will propagate to the next layer and become the input $\boldsymbol{x}^{(t+1)}$ of the next layer, (note that we omit the layer index for simplicity). $V_{th}$ is set to 0.5 in this work.

**Softmax in SNN.** In the general classification model, the final network output will be leveraged to compute the softmax and then to find the winner class. However, the network output is highly variant per input sample, and unlike ReLU activation, the output could be positive or negative. For this reason, we choose to only integrate the network output, and do not fire them across time, as did in recent practices [26, 27].

$$\boldsymbol{y}_{\text{net}} = \frac{1}{T} \sum_{i=1}^{T} \boldsymbol{c}_{\text{net}}^{(t)} = \frac{1}{T} \sum_{i=1}^{T} \mathbf{W} \boldsymbol{x}^{(t)}, \tag{3}$$

Then, we can compute the cross-entropy loss based on the ture label and $\text{Softmax}(\boldsymbol{y}_{\text{net}})$.

## 3.2 Time-dependent Batch Normalization

Batch Normalization (BN) [28] is used to reduce internal covariate shift and smooth the loss landscape [29] during the optimization of an ANN. It enables the stable training of ANN even with more than 100 layers on the large-scale dataset, e.g. ResNet-152 [30]. However, BN layers are not designed to normalize spatial-temporal data at first. And simply transplanting BN in SNN training will lead to undesired results. To solve this problem, Zheng *et al.* [26] propose time-dependent Batch Normalization (tdBN), which normalizes the data in both spatial and temporal paradigm. tdBN will gather the pre-synaptic input along the time dimension, denoted by $\mathbf{C} = [\boldsymbol{c}^{(1)}, \boldsymbol{c}^{(2)}, \dots, \boldsymbol{c}^{(T)}]$, and then normalize it by

$$\bar{\mathbf{C}} = \frac{\alpha V_{th}(\mathbf{C} - \mathbb{E}[\mathbf{C}])}{\sqrt{\mathbb{VAR}(\mathbf{C}) + \epsilon}} \qquad \qquad \text{\# normalize} \tag{4}$$

$$\hat{\mathbf{C}} = \gamma \bar{\mathbf{C}} + \beta, \qquad \qquad \text{\# affine transformation} \tag{5}$$

where $\mathbb{E}$ and $\mathbb{VAR}$ compute the mean and variance in channel dimension. $\gamma$ and $\beta$ are learnable parameters. Note that in tdBN the normalized tensor will multiply $\alpha V_{th}$, which can prevent over-fire and under-fire by choosing $\alpha$ wisely. After tdBN, the pre-synaptic input in each time step will be fed into Eq. (1). tdBN stabilizes the training of SNN and achieves good results. We also adopt this training technique in our framework. See [26] for more details of tdBN.

# 4 Methodology

## 4.1 Non-Differentiable Spike

One of the most notorious problems in SNN training is the non-differentiability of the firing function Eq. (2). The firing function could be viewed as a variant of sign function, whose derivative is identified as the Dirac delta function. This derivative is 0 almost everywhere except for the threshold that grows to infinity. Seemingly, directly using the Dirac delta function is not an ideal choice for gradient descent-based learning. To demonstrate this problem, let $L$ denote the task loss objective, we can compute the gradient of weights parameters using the spatial-temporal backpropagation (STBP):

$$\frac{\partial L}{\partial \mathbf{W}} = \sum_t \frac{\partial L}{\partial \boldsymbol{y}^{(t)}} \frac{\partial \boldsymbol{y}^{(t)}}{\partial \boldsymbol{u}^{(t),\text{pre}}} \frac{\partial \boldsymbol{u}^{(t),\text{pre}}}{\partial \boldsymbol{c}^{(t)}} \frac{\partial \boldsymbol{c}^{(t)}}{\partial \mathbf{W}}. \tag{6}$$

In above equation, $\frac{\partial \boldsymbol{y}^{(t)}}{\partial \boldsymbol{u}^{(t),\text{pre}}}$ is the gradient of firing function and is 0 almost everywhere, as aforementioned. As a consequence, the gradient descent ($\mathbf{W} \leftarrow \mathbf{W} - \eta \frac{\partial L}{\partial \mathbf{W}}$) either freezes weights or updates the weights to infinity.

To mitigate this problem, prior work seeks an alternative gradient (also known as surrogate gradient) to circumvent this problem. A popular surrogate gradient may refer to the rectangular function proposed in [4], given by

$$\frac{\partial \boldsymbol{y}^{(t)}}{\partial \boldsymbol{u}^{(t),\text{pre}}} = \frac{1}{a}\text{sign}\left(\left|\boldsymbol{u}^{(t),\text{pre}} - V_{th}\right| < \frac{a}{2}\right), \tag{7}$$

where $a$ is a hyper-parameter and usually set to 1. When $V_{th} = 0.5, a = 1$, the gradient is simplified to 1 if $0 \leq \boldsymbol{u}^{(t),\text{pre}} \leq 1$ otherwise 0. This is known as the Straight-Through Estimator in Bengio *et al*. [22] and is broadly applied in binary networks [31]. Another family of surrogate gradient may refer to triangular gradients [32], given by

$$\frac{\partial \boldsymbol{y}^{(t)}}{\partial \boldsymbol{u}^{(t),\text{pre}}} = \gamma \max\left(0, 1 - \left|\frac{\boldsymbol{u}^{(t),\text{pre}}}{V_{th}} - 1\right|\right). \tag{8}$$

This surrogate gradient is computed based on the distance between the potential and threshold. Despite various surrogate gradients achieve acceptable accuracy, there lacks a theoretical justification for its reason and effectiveness.

## 4.2 Rethinking Gradient-Descent in Spiking Neural Networks

Gradient-descent (GD) algorithm is a first-order iterative optimization method. Theoretically, for each iteration, GD aims to find the steepest direction for the descent that minimizes the task loss:

$$\min_{||\Delta\mathbf{W}||\leq\varepsilon} L(\mathbf{W} + \Delta\mathbf{W}) \approx \min_{||\Delta\mathbf{W}||\leq\varepsilon} L(\mathbf{W}) + \Delta\mathbf{W}\frac{\partial L}{\partial \mathbf{W}} \tag{9}$$

Note that the right-hand side of the minimization is approximated by the first-order Taylor expansion. Given that $L(\mathbf{W})$ is constant at the current iteration, therefore we only have to find the right $\Delta\mathbf{W}$ that minimizes the second term. Since the 2nd term is a linear in $\Delta\mathbf{W}$, it can be made as negative as we like by taking $\Delta\mathbf{W}$ large. Therefore the minimum is $\Delta\mathbf{W} = -\varepsilon\frac{\nabla_{\mathbf{W}}L}{||\nabla_{\mathbf{W}}L||}$, (where $\nabla_{\mathbf{W}}L = \frac{\partial L}{\partial \mathbf{W}}$).

Note that the above condition is only sufficient for $\varepsilon \to 0$ to correctly approximate the loss with Taylor expansion. However, $\varepsilon \to 0$ means $\Delta\mathbf{W} = -\varepsilon\frac{\nabla_{\mathbf{W}}L}{||\nabla_{\mathbf{W}}L||} \to 0$, but in practice GD updates the weights by a small step (determined by learning rate), which, indicates a mismatch between the practical implementation and the theoretical assumption. In other words, *gradient only inspects*

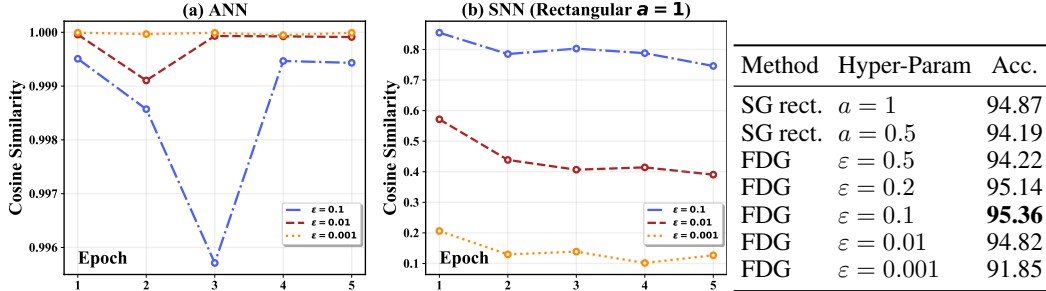

**Figure 2 & Table 1:** Left: The cosine similarity between the gradients used for update and the FDG. In (a) we show ANN gradients and in (b) we show SNN rectangular surrogate gradients. Right: Training SNN with finite difference gradient, given different $\varepsilon$.

*the change rate of loss around an infinitely small neighborhood, however, we are interested in the change rate around a tiny (but not infinitely small) area.* In the case of SNN, the gradient will always be $0^2$, yet for a tiny neighborhood the loss function will change, leading to non-zero gradients. To penetrate the non-differentiability of the firing function, we adopt a technique called *finite difference*. It estimates the gradient by evaluating the loss objective near the coordinate:

$$\hat{\nabla}_{\varepsilon,\boldsymbol{w}} L = \left( \frac{L(\boldsymbol{w}_1 + \varepsilon\boldsymbol{e}_1) - L(\boldsymbol{w}_1 - \varepsilon\boldsymbol{e}_1)}{2\varepsilon}, \ldots, \frac{L(\boldsymbol{w}_k + \varepsilon\boldsymbol{e}_k) - L(\boldsymbol{w}_k - \varepsilon\boldsymbol{e}_k)}{2\varepsilon} \right), \quad (10)$$

where $\boldsymbol{w}$ is the flattened vector of $\mathbf{W}$ and $\boldsymbol{e}_i$ refers to standard basis vector with 1 at $i$-th index. $\varepsilon$ is some small constant, e.g. 0.001. We call $\hat{\nabla}_{\varepsilon,\boldsymbol{w}} L$ as finite difference gradient (FDG). By the definition of gradient, we know FDG becomes the true gradient when $\varepsilon \to 0$. FDG can be a useful tool in GD-based learning, however, evaluating finite difference for each element in the weight vector is tedious, which greatly slows down the training process. Nevertheless, FDG still offers us a choice to evaluate the *gradient mismatch* problem quantitatively.

In order to illustrate this, we construct a toy problem: a simple 2-layer CNN on the MNIST dataset. The first layer is a 3x3 convolutional layer (output channel number is 6) with ReLU activation (or LIF spike activation) and average pooling for down-sampling. Then, we directly add a fully connected layer to predict the classification results. We train the toy model on MNIST for 5 epochs, with batch size 128 and learning rate 0.01 decayed to 0. In an effort to compute the FDG of the first layer, we have to perform $2 \times 1 \times 6 \times 3 \times 3 = 108$ times forward pass [3]. In contrast, one has to perform 22 million times forward passes on a ResNet-18 [30] to compute the FDG of all parameters.

**Experiment I: FDG in ANN.** We first train an ANN on MNIST and compare the disparity between real gradients and FDG. We use cosine similarity (i.e. $\text{cos\_sim}(\boldsymbol{x}_1, \boldsymbol{x}_2) = \frac{\boldsymbol{x}_1 \boldsymbol{x}_2}{||\boldsymbol{x}_1||_2 ||\boldsymbol{x}_2||_2}$) to determine the angle between these two gradient vectors. For each epoch, we compute the mean cosine similarity in the first 100 training iterations. The results are demonstrated in Fig. 2(a). The FDG and real gradients consistently have an extremely similar direction ($> 0.99$ cos\_sim) with $\varepsilon$ varying from 0.1 to 0.001. In conclusion, we show that (1) FDG is a valid signal and (2) the real gradients can effectively find the change rate within a small neighborhood in ANN optimization.

**Experiment II: FDG in SNN.** Then, we compare the cosine similarity between the surrogate gradients in SNN and the FDG. In Fig. 2(b), we can tell that surrogate gradients share similar gradient direction with FDG when $\varepsilon = 0.1$ (i.e. more than 0.8 cosine similarity). However, decreasing $\varepsilon$ will lead to lower cosine similarity. In the case of $\varepsilon = 0.001$, the angle between FDG and the surrogate gradients is nearly 90 degrees, (for comparison, in high-dimensional space, two arbitrary random Gaussian vectors are very likely to be orthogonal). This observation concludes that *surrogate gradient can predict the change rate of loss in a relatively larger neighborhood*, which justifies the soundness of surrogate gradients. However, SG fails to predict the FDG in small step size $\varepsilon$. To this end, we need to figure out whether a large step size is better or a small step size is better.

**Experiment III: Training SNN with FDG.** In SNN optimization, a natural question arises: *What $\varepsilon$ is optimal for gradient descent?* First, the $\varepsilon$ has to be small enough to compute the non-linearity

---

[2]Assuming the potential is not equivalent to $V_{th}$.

[3]According to Eq. (10), for each element we evaluate the loss twice, the first layer has 54 weight elements.

of the loss surface, meanwhile, it has to be large enough to ignore the noise from the rough step function. For this reason, we explore the optimal $\varepsilon$ by applying FDG in SNN optimization. The results are demonstrated in the Table 1. Note that all methods are trained on MNIST for only 10 epochs here. We show that the rectangular surrogate gradient with $a = 1$ reaches 94.9% accuracy. For FDG optimization, we evaluate $\varepsilon$ ranging from 0.001 to 0.5, and we find 0.1 is the best choice for optimizing SNN. The results confirm our assumption that $\varepsilon$ has to be neither too big nor too small. It also tells us using the instant change rate (very small $\varepsilon$) will lead to undesired performance degradation.

### 4.3 Differentiable Spike

In this section, we introduce the Dspike function, a novel surrogate gradient estimator, which can cover a large range of choices for SG. We choose the hyperbolic tangent function $\tanh(x) = \frac{e^x - e^{-x}}{e^x + e^{-x}}$ as our basic building function for gradient estimation:

$$\text{Dspike}(x) = \begin{cases} 1 & \text{if } x > 1 \\ a \cdot \tanh(b(x - c)) + d & \text{if } 0 \leq x \leq 1 \\ 0 & \text{if } x < 0 \end{cases} \tag{11}$$

When $x < 0$ or $x > 1$, we choose to clip $x$ as did in other practice. The vanilla $\tanh$ function maps $(-\infty, +\infty)$ to $(-1, 1)$, thus we have four parameters to control its shape to alter the function mapping: from $[-1, 1]$ to $[1, 1]$. We first use $c = V_{th} = 0.5$ to set the symmetry center to $(0.5, 0.5)$. Parameter $b(b > 0)$ is called the temperature factor, it controls the shape of the Dspike function. Parameter $a$ and $d$ can then be determined by setting $\text{Dspike}(0) = 0, \text{Dspike}(1) = 1$. As a result, we can rewrite the function as:

$$\text{Dspike}(x, b) = \frac{\tanh(b(x - 0.5)) + \tanh(b/2)}{2(\tanh(b/2))}, \text{if } 0 \leq x \leq 1. \tag{12}$$

Dspike function has the capability to adjust the shape of the function, leading to different surrogate gradient estimations. To demonstrate that, we have the following lemma:

**Lemma 4.1.** *By changing the temperature $b$, the shape of Dspike function can be altered. Specifically, we have $\lim_{b \to 0_+} \text{Dspike}(x) \to x$ and $\lim_{b \to +\infty} \text{Dspike}(x) \to \text{sign}(x - 0.5)$.*

*Proof.* We first prove the case $b \to 0_+$. In this case, applying Taylor expansion to the $\tanh$ we have:

$$\tanh(bx) \approx bx - \frac{b^3 x^3}{3} + \cdots + \frac{2^{(2n)}(2^{(2n)} - 1)B_{2n}(bx)^{2n-1}}{(2n)!}, \tag{13}$$

where $B_n$ is the $n$-th Bernoulli number. When $b \to 0$, we can safely ignore higher-order terms and only keep $bx$. Thus, we can rewrite Eq. (12) to $\text{Dspike}(x, b) = bx/b = x$. Then, we prove the case $b \to +\infty$. Since $\lim_{x \to +\infty} \tanh(x) \to 1$, we have:

$$\lim_{b \to +\infty} \text{Dspike}(x, b) \leftarrow \begin{cases} \frac{2*\tanh(+\infty)}{2*\tanh(+\infty)} = 1 & \text{if } x > 0.5 \\ \frac{\tanh(+\infty)}{2*\tanh(+\infty)} = 0.5 & \text{if } x = 0.5 \\ \frac{0}{2*\tanh(+\infty)} = 0 & \text{if } x < 0.5 \end{cases} \tag{14}$$

$\square$

With this temperature factor, our algorithm can simulate a wide range of surrogate gradients with different smoothness. We visualize the transition in Fig. 3 from both forward perspective and backward perspective. As aforementioned, surrogate gradient can mimic the FDG in a relatively large neighborhood. Our intuition is that the optimal surrogate gradient can be found by choosing temperature wisely. Thus our task becomes to optimize temperature and network parameters simultaneously during training. As did in former toy experiments (Sec. 4.2), we verify the cosine similarity between our Dspike gradients and the FDG at the beginning of each training epoch and update the temperature to better match the FDG. This algorithm is detailed in Algo. 1.

As abovementioned, evaluating FDG could be time-consuming, due to the two times evaluation of the final loss for each weight element. Therefore, we choose to only compute the FDG in the first layer for two reasons. First, the first layer has much fewer weights. In ResNet-18 (for CIFAR10 dataset), the first layer only has $3 \times 3 \times 3 \times 64 = 576$ elements. Second, the gradient propagation may accumulate the error in the former layers, which effectively measures the gradient mismatch.

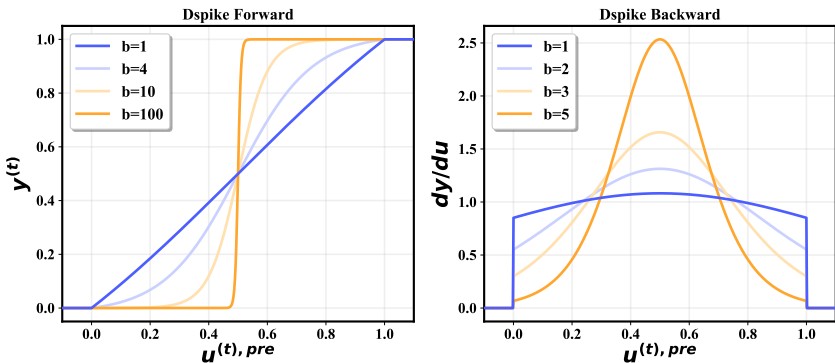

**Figure 3:** The forward plot (left) as well as the backward plot (right) of Dspike function. By changing temperature parameter we can manipulate the gradient estimation in SNN training.

---

**Algorithm 1:** Training SNN with Dspike gradient

---

**Input:** SNN to be trained; Training dataset, total training epoch $E$, total training iteration in one epoch $I$, FDG step size $\varepsilon$, temperature update step size $\Delta b$, initialized temperature $b^0$.

**for** *all $e = 1, 2, \ldots, E$-th epoch* **do**

    Collect input data and labels, computing the FDG ($\hat{\nabla}_{\varepsilon, \boldsymbol{w}} L$) of the first layer using $\varepsilon$;

    **for** *all $b$ in $\{b^{e-1}, b^{e-1} + \Delta b, b^{e-1} - \Delta b\}$* **do**

        Compute Dsipke surrogate gradient $\nabla_{b, \boldsymbol{w}} L$ and $\cos\_\text{sim}(\hat{\nabla}_{\varepsilon, \boldsymbol{w}} L, \nabla_{b, \boldsymbol{w}} L)$

    Find the optimal temperature with the highest cosine similarity and update it to $b^e$;

    **for** *all $i = 1, 2, \ldots, I$-iteration* **do**

        Get training data and labels, compute Dspike surrogate gradient using optimal $b$;

        Descend loss function and update weights;

**return** trained SNN.

---

### 4.4 Time Inheritance Training

We also introduce a novel training pipeline that can effectively reduce the time for training multiple SNNs. Unlike ANN-SNN conversion, direct training SNN is performed with a fixed time step $T$. As a result, we must learn a new SNN from scratch if we decide to use another different $T$. We propose Time Inheritance Training (TIT) pipeline to ameliorate this problem. First, we train an SNN to converge with high $T$, e.g. 20. Then, we use this well-trained model to initialize an SNN with lower $T$, say $T = 10$. Consequently, the initialized model has exactly the same output as the former model in the first 10 time steps, enabling better initialization and faster convergence. In other words, TIT progressively reduces the time step finetuning from a higher time step SNN first. This idea resembles the curriculum learning proposed in Bengio *et al.* [33], which suggests learning the easy task first. Armed with TIT, we can reduce the training time of subsequent models by 75%, while sharing the same task performances.

## 5 Experiments

In this section, we report experimental results for our Dspike algorithm on CIFAR [34] and ImageNet dataset [35]. For static image dataset, we encode the images to spike using the first layer in the SNN, as adopted in recent works [27]. For the neuromorphic image dataset, the images are already converted to 0/1 spike format. We also replace the max-pooling layer with the average-pooling layer in the network architecture. The detailed implementation is described in each subsection.

### 5.1 Network Architecture Choice

For CIFAR series experiments. We mainly adopt the ResNet-family architectures [30]. Unlike traditional ResNet structure [30], we use a modified version with 3 stem layers and avg-downsample

**Table 2:** Training Spiking Neural Networks on CIFAR10 and CIFAR100.

| Dataset | Method | Type | Architecture | Time Step | Accuracy |
|---|---|---|---|---|---|
| CIFAR10 | SpikeNorm [10] | ANN-SNN conversion | VGG-16 | 2500 | 91.55 |
| | Hybrid-Train [40] | Hybrid | VGG-16 | 200 | 92.02 |
| | Spike-based BP [41] | SNN training | ResNet-11 | 100 | 90.95 |
| | STBP [4] | SNN training | 5Conv, 2FC | 12 | 90.53 |
| | TSSL-BP [42] | SNN training | 5Conv, 2FC | 5 | 91.41 |
| | Diet-SNN [27] | SNN training | VGG-16 | 5 | 92.70 |
| | STBP-tdBN [26] | SNN training | ResNet-19 | 6 | 93.16 |
| | | | | 4 | 92.92 |
| | | | | 2 | 92.34 |
| | Dspike | SNN training | ResNet-18 | 6 | **94.25±0.07** |
| | | | | 4 | **93.66±0.05** |
| | | | | 2 | **93.13±0.07** |
| CIFAR100 | BinarySNN [43] | BNN-SNN conversion | VGG-15 | 62 | 63.20 |
| | Diet-SNN [27] | SNN training | VGG-16 | 5 | 69.67 |
| | Dspike | SNN training | ResNet-18 | 6 | **74.24±0.10** |
| | | | | 4 | **73.35±0.14** |
| | | | | 2 | **71.68±0.12** |
| CIFAR10-DVS | Rollout [44] | Streaming roll out ANN | DenseNet | 10 | 66.8 |
| | STBP-tdBN [26] | SNN training | ResNet-19 | 10 | 67.8 |
| | Dspike | SNN training | ResNet-18 | 10 | **75.4±0.05** |

block, as introduced in [36]. Our ResNet-18 architectures contains 4 stages with 2 basic residual block in each stage. Our initial channel is set to 64 and doubled in each stage. In comparison with [26], they use ResNet-19 with 3 stages, with 3 blocks in the first 2 stages and 2 blocks in the last stage. However, they choose 128 channel and they has larger receptive field than our model. In general, their ResNet-19 has ∼10 times more operations than our ResNet-18. For ImageNet, we use the same VGG-16 as used in [27] and the same ResNet-34 as used in [26].

## 5.2 CIFAR

The CIFAR10 (100) dataset contains 50K training and 10K test images with 32×32 pixels. We use AutoAugment [37] and Cutout [38] for data augmentation. We use ResNet-18 architecture for running experiments. We train the corresponding ANN first and use it to initialize the first SNN with ($T = 6$), then we use Time Inheritance Training to gradually reduce the time step to 2. For FDG computation, we use the best practice $\varepsilon = 0.1$ in the toy experiments. The temperature is initialized to 1, and we set the update step size to $\Delta b = 0.2$. We adopt SGD optimizer with 0.9 momentum. In the first round of TIT, we train the model for 300 epochs with a learning rate of 0.01 cosine decayed [39] to 0. In the rest rounds of TIT, we only train the model for 50 epochs with a learning rate of 0.004. Weight decay (L2 regularization) is set to 0.0001. We run the model with 4 GTX 1080Tis. For each run, we report the mean accuracy as well as the standard deviation with 3 trials.

We summarize the overall results in Table 2. On CIFAR10, our work can outperform state-of-the-art by 1.09% accuracy when $T = 6$. It is also worth noting that the ResNet-19 used in [26] has 2x channel numbers and a larger feature-map, translating to 10x computation cost. Nevertheless, our ResNet-18 with Dspike still achieves higher accuracy. Notably, our ResNet-18 ($T = 2, 4$) is trained with TIT for only 50 epochs, yet still reaches a new state-of-the-art. We also conduct an ablation study: training a 2 time step spiking ResNet-18 without TIT for 300 epochs. The accuracy is only 93.13%, which is subpar with the TIT method. We also verify Dspike on CIFAR100, with the same training pipeline here and we find our model can achieve much better performances than Diet-SNN [27].

To further demonstrate the effectiveness of our method, we apply the Dspike function on the event-stream dataset CIFAR10-DVS [45]. CIFAR10-DVS contains 10k 128×128 images converted from CIFAR10. We split the dataset into 9k training images and 1k test images following the prior practice [26]. For data pre-processing and augmentation, we first resize the training image frames to 48×48, then apply random horizontal flip. Additionally, we apply random roll within 5 pixels. The test images are directly resized to 48×48. Training hyper-parameters are kept the same with

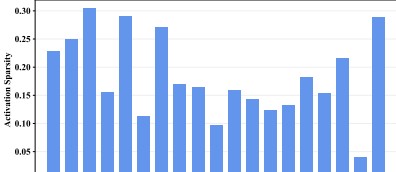

| Method | Model | Acc. | #Add. | #Mult. | Energy |
|---|---|---|---|---|---|
| ANN [26] | Res-19 | N/A | 2285M | 2285M | 12.6$J$ |
| ANN (ours) | Res-18 | 95.46 | 187M | 187M | 1.03$J$ |
| SNN [27](T=5) | Res-20 | 92.70 | 142M | 8.80M | 168$mJ$ |
| SNN [26](T=2) | Res-19 | 92.34 | 360M | 6.8M | 355$mJ$ |
| SNN (ours, T=2) | Res-18 | 93.13 | 71.2M | 3.52M | **80.3**$mJ$ |

**Figure 4 & Table 3:** Left: Activation sparsity for each LIF layer. The x-axis denotes the layer index while the y-axis denotes the activation sparsity. Right: The computation operations count and the energy consumption comparison between ANN and SNN for each run, we select SNN from existing literature.

**Table 4:** Training Spiking Neural Networks on ImageNet.

| Method | Type | Architecture | Time Step | Accuracy |
|---|---|---|---|---|
| SpikeNorm [10] | ANN-SNN conversion | ResNet-34 | 2500 | 65.47 |
| Hybrid-Train [40] | Hybrid | ResNet-34 | 250 | 61.48 |
| Hybrid-Train [41] | Hybrid | VGG-16 | 250 | 65.19 |
| STBP-tdBN [4] | SNN training | ResNet-34 | 6 | 63.72 |
| Diet-SNN [27] | SNN training | VGG-16 | 5 | 69.00 |
| Dspike | SNN training | ResNet-34 | 6 | **68.19** |
| Dspike | SNN training | VGG-16 | 5 | **71.24** |

the CIFAR10 experiments. Our proposed method can achieve 75.4% averaged top-1 accuracy on CIFAR10-DVS, with 7.6% absolute accuracy improvement compared with the existing state of the art.

## 5.3 ImageNet

In this section we verify our algorithm on the ImageNet dataset. ImageNet dataset has more than 1250k training images and 50k test images. We use the standard pre-processing and augmentation for training data [30]. The test data is directly centered cropped to 224×224. We train ResNet-34 and VGG-16 on ImageNet using the same architecture in the CIFAR experiments. Each network is trained for 100 epochs with SGD optimizer with 0.9 momentum. The learning rate is set to 0.04 and weight decay is set to 0.00004. We train each model on 32 GTX 1080Tis with batch size of 10 per GPU. Following [26], to avoid the biased statistics, we synchronize the batch mean and variance across each device. Results are demonstrated in Table 4, Dspike training obtain 4.5% improvement compared to STBP-tdBN [26] and 2.2% improvement compared to Diet-SNN [27].

## 5.4 Sparsity, Efficiency, and Energy Cost

In this section, we measure the sparsity and efficient computation as well as the energy consumption of SNN. For ANN, the dot-product performs multiply-accumulate (MAC) operation, which is composed of an addition and a multiplication. In the case of SNN, its event-based operation and binary activation enable multiplication-free (except for the first rate-encoding layer) inference. Only one addition is needed if a spike is activated. In Fig. 4 we visualize the mean sparsity of activation of our spiking ResNet-18 on the whole CIFAR10 test set. We can see that each LIF layer has no more than 32% sparsity rate, which greatly saves energy. We also measure the operation number in Table 3. The addition count is calculated by $s * T * A$, where $s$ is the mean sparsity, $T$ is the time step and $A$ is the addition number in ANN. For multiplication in SNN, we set it to the FLOPs of the first layer and scale it by $T$. Our spiking ResNet-18 with 2 time-step has the lowest operation need for the inference. Then, we measure the energy consumption. Following [27], the energy is measured in 45nm CMOS technology. The MAC operation in ANN costs $4.6pJ$ energy and the accumulation in SNN costs $0.9pJ$ energy. Based on this, we compute the energy cost and put it in the table. Our network only costs $80.3mJ$ for a single forward, and consumes $12.8\times$ lower energy compared with ANN.

## 5.5 Ablation Study

Our experiments use the same $\varepsilon$ for dataset and network architectures. In this section, we use different value for $\varepsilon$ and compare their results. The results is included in Table 5. Generally, we find a low temperature will decrease the performance, as we explained in Sec. 4.2, the desired optimization algorithm should ensure each update will decrease the loss function. Thus, the instantaneous derivative in SNN cannot measure such curvature information since the instantaneous

**Table 5:** Accuracy comparison given different $\varepsilon$.

| Dataset | $\varepsilon = 0.01$ | $\varepsilon = 0.05$ | $\varepsilon = 0.1$ | $\varepsilon = 0.2$ |
|---|---|---|---|---|
| CIFAR10 | 91.78 | 93.11 | **94.07** | 93.84 |
| CIFAR100 | 70.97 | 72.19 | 74.31 | **74.57** |

derivative is always 0. Therefore, we find a large epsilon can help improve the performance. Indeed, the optimal $\varepsilon$ requires tuning, but our empirical results show that 0.1 can have good results.

## 5.6 Visualization

In this section, we provide further experimental visualization. In Fig. 5, we plot the temperature curve for CIFAR10-DVS experiments. The spiking ResNet-18 was trained for 300 epochs with cosine learning rate decay. We initialize $b$ to 1.0 and use $\Delta b$=0.2 to optimize both temperature for gradient calculation and the weights parameters for network learning. For the initial 25 epochs, the temperature remains less-than-1. We think this is primarily due to the complex loss landscape at the starting stage of the training. During the 20 to 80 epoch, the temperature continues to be increasing. In this stage of learning the FDG desires to learn a sharp, narrow gradient with low bias but high variance. We also find the accuracy curve becomes more unstable in this stage of the training, as a consequence of the sharp gradient. After 80 epochs, the temperature becomes decreasing. Although the decreasing process is not aggressive as the former increasing stage, the trend is still obvious. From 80-250, the temperature decreases from 4.0 to 1.6 with some fluctuations. Then, at the end of the training stage, the temperature again becomes increasing.

The intuition behind this tuning mechanism may remain unexplainable right now, however, we think it is the intrinsic nature of the gradient estimation in SNN. Through FDG, we are able to optimize the gradient calculation along with the weights learning.

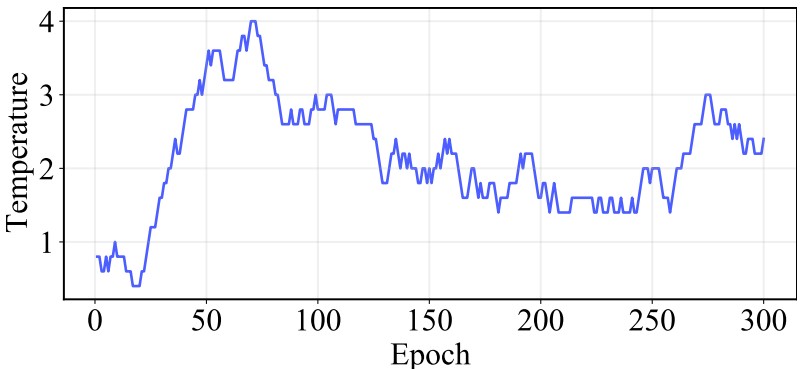

**Figure 5:** The evolutution of temperature $b$ during optimization.

## 6 Discussion

In this work, we inspect the training process of SNN when using the surrogate gradient. The proposed finite difference gradient and Dspike function can be assembled with other optimization methods as a fundamental module of gradient estimation for training SNN. This suggests that the performance of SNN can be further increased when more sophisticated training approaches are integrated with the currently proposed methods in the future. Although the improvement on model performance is very significant, be that as it may, our work is still an extension to the conventional SG. How to efficiently find the best FDG gradient to directly deal with the non-linearity of spiking function still remains a problem. In addition, we only consider the LIF neurons for the analyses here, the efficiency of the proposed approaches on other types of spiking neurons still needs examination in practice.

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
