# OpenReview forum: "Differentiable Spike: Rethinking Gradient-Descent for Training Spiking Neural Networks"
_NeurIPS.cc/2021/Conference — NeurIPS 2021 Poster_

### Official Review · Reviewer_RWyX · 2021-06-25

**Rating:** 5
**Confidence:** 5

**Summary:**

Summary.

This paper attempts to facilitate the surrogate gradient descent in training SNNs. The proposed FDG quantitatively analyzes the gradients in the training procedure. And then, the authors propose a Dspike function, which is claimed as a proper searching function for the optimal surrogate gradients.  Numerical experiments demonstrate the superiority of the proposed models to several contenders in CIFAR and ImageNet data sets.

Contributions.

I have to admit that it's hard to understand and follow the objective and soundness of this paper. At first glance at the Tables in Section 5, the proposed methods achieve a commendable result compared to other SNN models. However, I can't entirely agree with the author's claimed contributions, especially the first and second ones.

As claimed that "through which (FDG) we are able to not only explain why it (surrogate gradient) works well but also indicate how we can improve its (gradient) adaptivity to data through training", so what are the answers of why and how? In detail,

[1] The derivation of FGD can be traced back to the directional derivative and difference in mathematics. It is reasonable to use Eq. (10) to approximate the instantaneous derivative at a certain point in ANNs, which have continuous and differentiable activations. The high-similarity curves in experiment $\mathrm{I}$ verify my conjectures. It's a good point.

[2] However, the finite difference technique is not suitable for estimating the derivative of holder-continuous and non-differentiable functions. There may be a large gap between the instantaneous and average derivatives. Thus, the similarity between FDG and surrogate gradients in SNNs cannot support the "soundness of surrogate gradients" in Line 175 due to the imprecision of FDG.

[3] Further, how to conclude that "This observation concludes that surrogate gradient can predict the change rate of loss in a relatively larger neighborhood" between Lines 175-176? Doesn't dissimilarity mean invalidity? I hope the authors can clarify this point.

[4] My conjecture in [2] can be verified by Figure 2. At the initial state that epoch = 0 or 1, parameters $\mathbf{W}$ may be far away from the optimal solution. In this case, FDG is not similar to the surrogate gradient if employing small $\epsilon$. As the setting in the experiment that "weight decay (L2 regularization) is set to 0.0001", the parameters may converge to the optimal local minima as weights decay to 0.0001. Thus, FDG still works, although $\epsilon$ is small. So it may be a practical trick but not holds as the authors theoretically claimed, right?

About the claim that "optimal surrogate gradient can be found by choosing temperature wisely" in Lines 210-211. In other words, the Dspike function is effective since optimal surrogate gradient can be found by choosing temperature wisely. How to evaluate this point? I would have appreciated more clarity on the experimental results not as omitted by "As did in former toy experiments (Sec. 4.2), we verify the cosine similarity between our Dspike gradients and the FDG at the beginning of each training epoch and update the  temperature to better match the FDG."

In summary, I find the interests of this work and the proposed model may be effective in practice. However, this conclusion should be testified either theoretically or experimentally. Of course, this paper still needs improvement in these two aspects. So the current version is improper for acceptance.

**Limitations And Societal Impact:**

As mentioned above.

**Main Review:**

Clarity and Quality.

[1] The formalization and theoretical derivation of this paper are clear.

[2] The conclusion and proof of Lemma 4.1 are valid and correct. Unfortunately, this lemma only describes the properties of Dspike function. Can the function set of Dspike functions with learnable parameters contain the surrogate gradients obtained by the common-used approaches, such as Eqs. (7) or (8)?

[3] What's the computational complexity of FDG? It is better to show the number of paras, computational overhead, and storage costs.

[4] As shown in Table 2, adding TIT will reduce the performance of Dspike. TIT is not the contribution of this work; why do this here?

[5] The experimental results in Table 2 are not promising. Taking the results in CIFAR-100 as an example, both architecture and Dspike functions are different. So how is the improved accuracy attributed to the power of the Dspike function with FDG? In my mind, ablation experiments are desired. Besides, the comparative models are so scant.


**Time Spent Reviewing:**

3 hours

---

> ### Author Response · Authors · 2021-08-11
> **Response to Reviewer RWyx**
>
> Thank you for your constructive feedback. Please see our detailed reply below.
>
> Q1: However, the finite difference technique is not suitable for estimating the derivative of holder-continuous and non-differentiable functions. There may be a large gap between the instantaneous and average derivatives. Thus, the similarity between FDG and surrogate gradients in SNNs cannot support the "soundness of surrogate gradients" in Line 175 due to the imprecision of FDG.
>
> A1: That is correct, the FDG cannot measure the instantaneous derivatives (unless with a tiny $\varepsilon$). However, in Line 142-147, we argue that the optimization algorithm should measure the neighborhood curvature rather than the instantaneous curvature. Especially in SNN, we think the instantaneous curvature (which is the true gradient) is not meaningful since it is always zero. However, when we change the weights, the loss objective would always change (unlike the true gradient descent where zero gradients won't update the weights at all). Therefore it is important to search for a direction that measures how loss function will change. In this work, we use a finite difference method to predict how the loss function changes. Our experiment $II$ shows that surrogate gradients have similar directions with FDG$_{\varepsilon=0.1}$. This demonstrates that surrogate gradients can somewhat capture the loss change rate in the relatively large neighborhood area, which is useful for SNN optimization.
>
> ------
>
> Q2: Further, how to conclude that "This observation concludes that surrogate gradient can predict the change rate of loss in a relatively larger neighborhood" between Lines 175-176? Doesn't dissimilarity mean invalidity? I hope the authors can clarify this point.
>
> A2: In FDG, the hyper-parameter decides how far the loss is evaluated. With $\varepsilon\rightarrow 0$, the FDG will get close to the true gradient, i.e. instantaneous derivative. In our experiment $II$, we find surrogate gradients get similar directions with larger epsilon, this result indicates that surrogate gradients can predict the change rate of loss in a relatively larger neighborhood.
>
> ------
>
> Q3: My conjecture in [2] can be verified by Figure 2. At the initial state that epoch = 0 or 1, parameters W may be far away from the optimal solution. In this case, FDG is not similar to the surrogate gradient if employing small ϵ. As the setting in the experiment that "weight decay (L2 regularization) is set to 0.0001", the parameters may converge to the optimal local minima as weights decay to 0.0001. Thus, FDG still works, although ϵ is small. So it may be a practical trick but not holds as the authors theoretically claimed, right?
>
> A3: Thanks for your comment. Our cosine similarity calculation does not include the L2 regularization. This is because the L2 regularization is performed in `optimizer.step` in pytorch package while the gradient calculation is performed in `loss.backward`. I am not sure why the L2 regularization is related to the analysis, could you please elaborate on it?
> As shown in Fig. 2, we do not find the cosine similarity has changed a lot during the training. Instead, we find the choice of epsilon is crucial. As explained in A1, surrogate gradients have similar directions with FDG$_{\varepsilon=0.1}$, thus we conclude that surrogate gradients can measure the change rate of loss in a relatively large neighborhood.
>
> -----------
>
> Q4: About the claim that "optimal surrogate gradient can be found by choosing temperature wisely" in Lines 210-211. In other words, the Dspike function is effective since optimal surrogate gradient can be found by choosing temperature wisely. How to evaluate this point?
>
> A4: Thanks for the comment. We have to admit that one may never know if his choice of surrogate gradients is generalizable to optimal situations. We should revise this sentence to “A good surrogate gradient can be found by choosing temperature wisely”. Sorry for the overclaiming. Our design of this Dspike function is to cover the smooth function as much as possible. We hypothesize that our Dspike function is diverse enough to have a good approximation. Furthermore, we would like to clarify our core contribution is to jointly optimize gradient estimation and the weight parameters.
>
> -----------
>
> Q5: I would have appreciated more clarity on the experimental results not as omitted by "As did in former toy experiments (Sec. 4.2), we verify the cosine similarity between our Dspike gradients and the FDG at the beginning of each training epoch and update the temperature to better match the FDG."
>
> A5: The detailed algorithm process is demonstrated in Algorithm 1 (page 7). Specifically, we jointly optimize the temperature parameter and the network weights parameters. In each epoch, we compute FDG and the several surrogate gradients and choose the best matched (highest cosine similarity) temperature to adjust surrogate gradients.
>
> -----------
>
> Q6: Can the function set of Dspike functions with learnable parameters contain the surrogate gradients obtained by the commonly-used approaches, such as Eqs. (7) or (8)?
>
> A6: Yes, the rectangular surrogate gradients are a special case of our Dspike function. When b is set to near 0, the function becomes linear and has rectangular surrogate gradients. As for the triangular gradients, its function is a polynomial function while ours is an exponential function. Still, the exponential function with large b shares a similar shape with the polynomial function.
>
> -----------
>
> Q7: What's the computational complexity of FDG? It is better to show the number of paras, computational overhead, and storage costs.
>
> A7: This is a good question. In our CIFAR experiments. Computing FDG nearly costs the same time as weight training. Therefore, applying FDG may double the training time than the original method. FDG method does not include other memory overhead since we only use forward computation which means the memory cost is lower than the normal training process. We admit that FDG is not perfect by now because it has more training costs. Our contribution is to offer another perspective in the SNN gradient descent algorithms. Thus we would like to leave the computational parts in future work. We will add the comparison of training costs in the next version of the paper.
>
> -----------
>
> Q8: As shown in Table 2, adding TIT will reduce the performance of Dspike. TIT is not the contribution of this work; why do this here?
>
> A8: This is not true. We do not report the accuracy of none TIT training in Table 2. In practice, we find TIT has the same accuracy as the normal method (the accuracy difference is within 0.1%), while TIT can significantly reduce the training time. For example, 200 epochs can be reduced to 50 epochs.  Furthermore, TIT is our contribution, although it is a minor one. This technique can reduce the training time significantly.
>
> -----------
>
> Q9: The experimental results in Table 2 are not promising. Taking the results in CIFAR-100 as an example, both architecture and Dspike functions are different. So how is the improved accuracy attributed to the power of the Dspike function with FDG? In my mind, ablation experiments are desired. Besides, the comparative models are so scant.
>
> A9: Please see our ablation experiments in general response. Thank you!

---

> ### Author Response · Authors · 2021-09-02
> **Response to Reviewer RWyx -- A clarification on our contribution**
>
> Thanks for your constructive discussion on FDG and your time in reading our paper and rebuttal. Here I want to clarify a bit more on the contribution and completeness of our work.
>
> 1. Our main contribution is NOT to create the FDG for general purposes but to apply it to solve the degeneration of representivity in training SNN caused by the discontinuity of gradients. This improvement has been demonstrated through our results much beyond the existing SOTA. This flexible gradient is actually unique for SNN and thus very inspiring to build a pool of representations different from the existing ANNs. We agree that a more detailed discussion on FDG could expand its utility in a broad range of applications, but that's off the current scope of improving the training efficiency of SNN. The current submission is not perfect, but the content and logic are fairly complete given its target.
>
>
> 2. In terms of your current concerns on FDG techniques and ablation study, we really appreciate your patience here if you can spare some time to check our response. Indeed, most of them have already been contained in the current context.
>
>
> Based on these two points, we really hope that you can re-consider your judgment here. We are happy to discuss further if you have any concerns or questions.

---

### Official Review · Reviewer_uEo9 · 2021-06-27

**Rating:** 6
**Confidence:** 4

**Summary:**

This paper proposes a new method called Differentiable Spike (Dspike) to compute the surrogate gradient for backpropagation of SNNs. In addition, an online optimization method is proposed to optimize the hyperparameter of Dspike.

**Limitations And Societal Impact:**

1. The main issue of this paper is novelty. Overall, this main idea of this paper is a surrogate function with the dynamic hyperparameter tuning method to enable wide range of smoothness. However, it is still in the framework of surrogate gradient method and I cannot view it as a significant improvement over previous works. This work reports a good results on several challenging datasets like CIFAR10-DVS, CIFAR100 and ImageNet. I appreciate the authors' efforts in the experiments and I think the performances reported in this paper are better than state-of-the-art results. However, in my opinion, this is more about engineering works for running experiments and hyperparameters tuning.

2. For the FDG computation of the experiments, the authors set epsilon to 0.1 which is empirically coming from the toy experiments on MNIST. However, I think this parameter should depend on datasets and cannot find a universal optimal choice. Can the authors comment on why the epsilon is fixed in the experiments?

3. In table 2, I hope the authors can compare the proposed method with existing works using the same network size or similar number of tunable parameters. Thus, we can understand the improvement brought by the proposed method.

4. For section 5.3, I hope the author can analyze the complexity of the proposed method compared to the existing methods. For example, compared with the STBP method, how much improvement the proposed method can achieve and how much additional cost do the dynamical temperature tuning and time inheritance training method require?

**Main Review:**

Strength: I appreciate that this paper demonstrates results on more complicated datasets like CIFAR 100 and ImageNet.

Weakness: The proposed method is still in the frame of surrogate gradient.

Please see the comments below for details.

**Time Spent Reviewing:**

4 hrs

---

> ### Author Response · Authors · 2021-08-11
> **Response to Reviewer uEo9**
>
> Q1: The main issue of this paper is novelty. Overall, this main idea of this paper is a surrogate function with the dynamic hyperparameter tuning method to enable wide range of smoothness. However, it is still in the framework of surrogate gradient method and I cannot view it as a significant improvement over previous works. This work reports a good results on several challenging datasets like CIFAR10-DVS, CIFAR100 and ImageNet. I appreciate the authors' efforts in the experiments and I think the performances reported in this paper are better than state-of-the-art results. However, in my opinion, this is more about engineering works for running experiments and hyperparameters tuning.
>
> A1: Overall, our most significant contribution to the SNN field is we recognize and demonstrate that the discontinuity in gradient can be utilized to improve the representativity of SNN thus increase SNN’s prediction accuracy. This simple and effective discovery adds unique adaptivity of the gradients of SNN to the data through training beyond the approximation to ANN. It brings novel perspectives of understanding the representing power of SNN that allows future works to use as a plug-in module. In terms of surrogate gradients, it is true that our method is still in the frame of surrogate gradients. In the general sense, all iterative methods for SNN training can be viewed as certain types of surrogate gradients since its gradient cannot be accurately computed. In addition to this conceptual novelty, our ablation study also confirms that our excellent performance beyond SOTA is not simply a result of engineering works but due to the insight into the uniqueness in SNN. In addition, **we would like to clarify that existing surrogate gradients are also hyperparameter tuning works. Eq. 7 and Eq. 8 both have hyperparameters to control their shapes.**
>
> --------
>
> Q2: For the FDG computation of the experiments, the authors set epsilon to 0.1 which is empirically coming from the toy experiments on MNIST. However, I think this parameter should depend on datasets and cannot find a universal optimal choice. Can the authors comment on why the epsilon is fixed in the experiments?
>
> A2: Thanks for your constructive comments. We agree that the optimal parameter may depend on the architecture as well as the dataset. In our general response, we include an experimental comparison on different $\varepsilon$. Still, we find a relatively large value of the $\varepsilon$ is more effective. This result corresponds to our findings back in Sect 4.2. Since the instantaneous derivative in SNN cannot measure the loss curvature information (which would always be 0), we think a large step would generally have better results. We also conducted a 0.2 epsilon experiment and found similar performance. Overall, we think finding the best value can improve performance, but 0.1 generally yields good performance.
>
> ---------
>
> Q3: In table 2, I hope the authors can compare the proposed method with existing works using the same network size or similar number of tunable parameters. Thus, we can understand the improvement brought by the proposed method.
>
> A3: Thank you for your suggestion. We use Res-19 which is the same in the STBP-tdBN work to conduct an ablation study. Results are shown in the General Response section. Note that we omit the VGG-network comparison since the SOTA method Diet-SNN used the same architecture as ours.
>
> --------
>
> Q4: For section 5.3, I hope the author can analyze the complexity of the proposed method compared to the existing methods. For example, compared with the STBP method, how much improvement the proposed method can achieve and how much additional cost do the dynamical temperature tuning and time inheritance training method require?
>
> A4: It's a good question. For FDG-based gradient optimization, we found generally our method will double the training time. 50% for computing the FDG (requires ~100 times forward computation) and 50% for backpropagation using Dspike and weights update. FDG method does not include other memory overhead since we only use forward computation which means the memory cost is lower than the normal training process. We admit that FDG is not perfect by now because it has more training costs. Our contribution is to offer another perspective in the SNN gradient descent algorithms. Thus we would like to leave the computational parts in future work and improve this method. Fortunately, we also have TIT training algorithms, which can significantly reduce the training time. As we explained, TIT utilizes a higher T trained model as initialization to lower T models which can dramatically speed up the training process. With TIT, we can only train an SNN in 50 epochs rather than 200 epochs in normal training.

---

### Official Review · Reviewer_uSGb · 2021-07-18

**Rating:** 5
**Confidence:** 3

**Summary:**

This paper studies the differentiable training problem of spiking neural network (SNN), which is a promising alternative to the widely used ANN. The authors first identify the problem of the current surrogate gradient of SNN and propose a new surrogate gradient estimator. I really like the author's research method of thinking about problems from experiments, but the result of the experiment is just a tuning experiment of some parameters, and I did not see the author's insight into the existing SNN training problems. Furthermore, the Spike method proposed by the author lacks clear motivation. The proposed Dspike method is just a differentiable piecewise function, which does not have much methodological contribution. The author exaggerated the experimental settings and did not present the experimental results of neuromorphic images. In summary, I think this article does not meet the standards of the NeurIPS conference.

The author’s reply addressed most of my concerns. I agree that AdderNet is another line of work with different purpose. I can see that the authors are very thoughtful in the model design, however it is not well-described in the main paper. I have raised my rating by 1.

**Limitations And Societal Impact:**

Yes, the authors addressed the limitations and potential improvements.

**Main Review:**

Main Comments:
1.	AdderNet [a] is also a multiplication-free neural network architecture. What is the advantage and disadvantages of Spiking Neural Network?
2.	How did Time-dependent Batch Normalization (TdBN) relevant to this paper? What is the motivation for using time-dependent batch normalization? The authors argued that TdBN is designed to normalize spatial-temporal data (Line 103-104), Are there any experimental results that argument?
3.	What is the motivation of the Time Inheritance Training (TIT) proposed in Section 4.4?
4.	The authors argue that they conduct experiments on both static and dynamic datasets (Line 51), however, the authors only reported results on static datasets (CIFAR and ImageNet Sec 5.1 and Sec 5.2), and I didn’t find any results related to dynamic dataset (including Supplementary materials ). Where are the results on the dynamic dataset? In Line 235, the authors mentioned a “neuromorphic image dataset”, is that the dynamic dataset? Where are the experimental results on that dataset?


**Time Spent Reviewing:**

5

---

> ### Author Response · Authors · 2021-08-11
> **Response to Reviewer uSGb**
>
> Q1: AdderNet [a] is also a multiplication-free neural network architecture. What is the advantage and disadvantages of Spiking Neural Network?
>
> A1: Spiking Neural Networks are biology-inspired networks that are thought to exhibit favorable properties such as power consumption, inference speed, and event-driven input when deployed properly on neuromorphic hardware. It provides a different way of representing an object/input to the traditional ANNs. Its power in representing an object inspired by the creature’s brains is still an emerging field to explore. Currently, the main disadvantage of SNN is the lack of efficient training algorithms to achieve its optimum as expected. This is why we propose the algorithm in the current paper to push the frontier of the area. AdderNets are efficient networks that speed up the processing by trading multiplications to additions. In our understanding, its target is to improve the hardware efficiency rather than obtaining a different way of representation like SNN.
>
> ----------
>
> Q2: How did Time-dependent Batch Normalization (TdBN) relevant to this paper? What is the motivation for using time-dependent batch normalization? The authors argued that TdBN is designed to normalize spatial-temporal data (Line 103-104), Are there any experimental results for that argument?
>
> A2: Since the neurons in SNN emit spikes rather than real values, we cannot use the traditional batch normalization in the training step. TdBN is an approach that allows for Batch Normalization in training SNNs. It is proposed and discussed in [1] with rich experimental results to support its validity. We adopt it here to show that our method is compatible with other progressions in the field thus can be used in a plug-in manner to improve the SNN’s performance.
>
> -------------
>
> Q3: What is the motivation of the Time Inheritance Training (TIT) proposed in Section 4.4?
>
> A3: This method is proposed to reduce the training time while preserving the same performance of other trained from scratch methods. As we explained, TIT utilizes a higher T trained model as initialization to lower T models which can dramatically speed up the training process.
>
> -----------
>
> Q4: The authors argue that they conduct experiments on both static and dynamic datasets (Line 51), however, the authors only reported results on static datasets (CIFAR and ImageNet Sec 5.1 and Sec 5.2), and I didn’t find any results related to dynamic dataset (including Supplementary materials ). Where are the results on the dynamic dataset? In Line 235, the authors mentioned a “neuromorphic image dataset”, is that the dynamic dataset? Where are the experimental results on that dataset?
>
> A4: The neuromorphic image dataset is CIFAR10-DVS, we include the results in Table 2. Please check it again.
>
> [1] Zheng, Hanle, et al. "Going deeper with directly-trained larger spiking neural networks." arXiv preprint arXiv:2011.05280 (2020).

---

> ### Author Response · Authors · 2021-08-12
> **Response to your second run of rating**
>
> Thanks for your recognition of our thoughtfulness in the model design and your improvement on the rating.
>
> Considering the length of the submitted draft, we cannot elaborate on every point, but the main part we strengthened in the general response is already there and well-described. Here we provided the exact lines to your concerns and really appreciate it if you can kindly re-check such existing content and re-consider your ratings.
>
> 1. It was in our original submission rather than a post-hoc interpretation in the rebuttal that we identified our major methodological contribution as identifying the adaptivity in gradients. This is summarized in lines 46-48 as the first piece of our contribution.
>
> 2. In lines 164-188 for the experiments, we not only stated the engineering details but also provides explanations on why we set it up in the current way and how we can interpret the findings.
>
> 3. In terms of TIT, the related lines are 221-231. We also indicated that it was used for better initialization and convergence and pointed the motivation.
>
> 4. While we provided an additional ablation study in the rebuttal, the original submission already included the results for CIFAR10-DVS with very significant improvement over the existing SOTA.  This was shown in Table 2 on Page 8.
>
> Based on such existing content, we believe that our contribution has already been supported in the current submission. We understand the possibility of missing lines during the intensive review procedure, but we really hope that you can kindly spend some time checking the lines we refer to above and re-consider your rating. On the other hand, we do not mean that we already make everything clear, and we will continue to polish the description in the future version. Thank you again and hope to hear your further comments and suggestions.

---

### Official Review · Reviewer_Q9Mk · 2021-07-20

**Rating:** 8
**Confidence:** 4

**Summary:**

The authors outline a principled approach--finite difference gradients (FDG)--for choosing a surrogate gradient for spiking neural networks. FDG is demonstrated to approximate gradient descent in classic rate-based ANNs very well, approximate classic rectangular surrogate gradient function descent in SNNs for medium neighborhoods and thus theoretically justify the method, and be useful in finding an optimal neighborhood in which to approximate a gradient for SNNs.  This last step is then used to optimize a continuously varying surrogate spike gradient estimator (Dspike), modeled as a tanh approximation with the gain as a free parameter. Using the chosen Dspike function (which depends on the time step), the authors then demonstrate the appreciable accuracy and energy efficiency improvements of the resultant SNN.

**Limitations And Societal Impact:**

Yes

**Main Review:**

The authors' approach is original (to the reviewer's knowledge), intuitive, and of significant interest. The approach and results are well motivated, well written, and understandable, and appear to represent an appreciable advance in the field of SNNs.  However, some of the comparisons that have been made leave some room for questions as to the specific improvements that Dspike provides. Indeed, some of the improvements may be greater than suggested. Addressing this would be helpful for readers to better assess the impact of the authors' work.

In particular, Table 2 shows that Dspike achieves greater accuracy than all other surveyed methods.  However, the comparisons vary in important categories, in particular, architecture. Dspike may well do better than suggested compared to STBT-tdBN for both CIFAR 10 and CIFAR 10-DVS, since the former method utilized ResNet-18 and the latter ResNet-19.  Meanwhile, the comparisons to BinarySNN and Diet-SNN are less clear since they use VGG-15 and VGG-16, respectively. It would be helpful if, in these comparisons, the comparisons involved only a change of method and not architecture as well. Note that Table 4 does indeed allow such comparisons, and the resultant improvements are potentially somewhat more modest.

**Time Spent Reviewing:**

3

---

> ### Author Response · Authors · 2021-08-11
> **Response to Reviewer Q9Mk**
>
> Thank you for your positive feedback. Your concerns are the same with Reviewer uEo9 and RWyx. Therefore we put the ablation study in the General Response section. If you have further questions, please do not hesitate to reply to us. Thanks again.

---

### Author Response · Authors · 2021-08-11
**General Response**

We thank all reviews’ time and effort in reviewing our paper. While we will respond to the comments piece by piece, we would like to first provide a general response to the common issues and reemphasize our contribution to the SNN field on both methodological and experimental levels.

On the methodological level, one of the most important questions in training SNN is how to deal with the discontinuity of the spiking function. Many existing works focus on finding appropriate surrogate functions to compensate for the deficit in model representativity due to such discontinuity but ignore the possibility that the inexistence of gradients also leaves space for constructing more effective gradients. In this work, we recognize that such discontinuity is NOT simply a weakness, it actually allows for additional adaptivity of the gradients of SNN to the data through training. So, we conclude that the identification of this special mechanism for SNN is our most novel contribution to the SNN field since it distinguishes SNN from a degenerated version of discretized ANN in its representing power.

In terms of the experiments, we appreciate the reviewer’s suggestions for ablation studies. The reason for adopting some advanced training techniques from ANN is to show that the proposed approach is compatible with other advances in the field so that it can be applied in a plug-in manner to improve the model performance on complex datasets. This workable and sophisticated training pipeline is also our technical contribution to the SNN field, providing a followable benchmark for the future works of SNN. As stated by Reviewer Q9MK and agreed by all reviewers, our model’s improvement over the SOTA is very significant, especially for the CIFAR-DVS. In practice, it is unlikely that this level of improvement is simply from the parameter tuning rather than doing something right methodologically. We provide the results of ablation studies below to address the reviewer’s concerns and support our methodological contribution.

**Architecture Ablation Study**

For VGG experiments, our work, as well as the current SOTA method Diet-SNN, both use VGG-16 architecture. A prior work uses VGG-15, however, it achieves much lower results than ours, therefore we omit the comparison in VGG-net.
For ResNet evaluation, we here run our experiments on the same Res-19 architecture. The results are shown below. Since Res-19 is much larger than our Res-18 (they double the channel numbers), our Dspike can get higher accuracy than STBP-tdBN.

| Method    | Architecture | Time Step | CIFAR10 Accuracy | CIFAR10-DVS Accuracy |
|-----------|--------------|-----------|------------------|----------------------|
| STBP-tdBN | ResNet-19    | 6 / 10    | 93.16            | 67.8                 |
| Dspike    | ResNet-19    | 6 / 10    | 94.07            | 77.0                 |

**Temperature Experiments**

We agree that different datasets may have optimal temperature choices. We conduct an ablation study here using the Res-19:

| Dataset  | epsilon=0.01 | epsilon=0.05 | epsilon=0.1 | epsilon=0.2 |
|----------|--------------|--------------|-------------|-------------|
| CIFAR10  | 91.78        | 93.11        | **94.07**       | 93.84       |
| CIFAR100 | 70.97        | 72.19        | 74.31       | **74.57**       |

Generally, we find a low temperature will decrease the performance, as we explained in Sect 4.2, the desired optimization algorithm should ensure each update will decrease the loss function. Thus, the instantaneous derivative in SNN cannot measure such curvature information since the instantaneous derivative is always 0. Therefore, we find a large epsilon can help improve the performance. Indeed, the optimal epsilon requires tuning, but our empirical results show that 0.1 epsilon can have good results.

---

### Decision · Program_Chairs · 2021-09-27

**Decision:**

Accept (Poster)

**Comment:**

The authors present a method to train spiking neural networks (SNNs) using an adaptive surrogate gradient method.
Training of SNNs is difficult due to the well-known problem of non-existing derivative of the spiking neuron output. This is usually tackled using a surrogate gradient method. However, the functional form of the surrogate gradient is quite arbitrary.

The authors first analyze the training behavior of SNNs. They propose to use a parameterized function for the surrogate gradient (sg) and estimate its best actual shape based on a comparison with a finite-difference gradient. Then they use this sg for parameter updates. They test their method on challenging data sets such as CIFAR100 and ImageNet and show excellent performance.

The paper is very well written and clear. The adaptive surrogate gradient is a novel contribution that has potential to impact future SNN training methods. The experimental results are compelling.

Weaknesses and potential improvements:
The functional form of the adaptive surrogate gradient lacks motivation.
Reviewers have identified some unclear or over-stated statements. The authors should refine these statements accordingly.